# Novel Polymorphisms and Genetic Characteristics of the Shadow of Prion Protein Gene (*SPRN*) in Cats, Hosts of Feline Spongiform Encephalopathy

**DOI:** 10.3390/v14050981

**Published:** 2022-05-06

**Authors:** Yong-Chan Kim, Hyeon-Ho Kim, Kiwon Kim, An-Dang Kim, Byung-Hoon Jeong

**Affiliations:** 1Korea Zoonosis Research Institute, Jeonbuk National University, Iksan 54531, Jeonbuk, Korea; kych@jbnu.ac.kr (Y.-C.K.); khh7051@jbnu.ac.kr (H.-H.K.); 2Department of Bioactive Material Sciences and Institute for Molecular Biology and Genetics, Jeonbuk National University, Jeonju 54896, Jeonbuk, Korea; 3Haemalken Animal Hospital, Yangju 11492, Gyeonggi, Korea; kkw0075@hanmail.net; 4Cool-Pet Animal Hospital, Anyang 14066, Gyeonggi, Korea; kad7582@hanmail.net

**Keywords:** cats, prion, *PRNP*, *SPRN*, SNP, feline, FSE

## Abstract

Prion diseases are transmissible spongiform encephalopathies (TSEs) caused by pathogenic prion protein (PrP^Sc^) originating from normal prion protein (PrP^C^) and have been reported in several types of livestock and pets. Recent studies have reported that the shadow of prion protein (Sho) encoded by the shadow of prion protein gene (*SPRN*) interacts with prion protein (PrP) and accelerates prion diseases. In addition, genetic polymorphisms in the *SPRN* gene are related to susceptibility to prion diseases. However, genetic polymorphisms in the feline *SPRN* gene and structural characteristics of the Sho have not been investigated in cats, a major host of feline spongiform encephalopathy (FSE). We performed amplicon sequencing to identify feline *SPRN* polymorphisms in the 623 bp encompassing the open reading frame (ORF) and a small part of the 3′ untranslated region (UTR) of the *SPRN* gene. We analyzed the impact of feline *SPRN* polymorphisms on the secondary structure of *SPRN* mRNA using RNAsnp. In addition, to find feline-specific amino acids, we carried out multiple sequence alignments using ClustalW. Furthermore, we analyzed the N-terminal signal peptide and glycosylphosphatidylinositol (GPI)-anchor using SignalP and PredGPI, respectively. We identified three novel SNPs in the feline *SPRN* gene and did not find strong linkage disequilibrium (LD) among the three SNPs. We found four major haplotypes of the *SPRN* polymorphisms. Strong LD was not observed between *PRNP* and *SPRN* polymorphisms. In addition, we found alterations in the secondary structure and minimum free energy of the mRNA according to the haplotypes in the *SPRN* polymorphisms. Furthermore, we found four feline-specific amino acids in the feline Sho using multiple sequence alignments among several species. Lastly, the N-terminal signal sequence and cutting site of the Sho protein of cats showed similarity with those of other species. However, the feline Sho protein exhibited the shortest signal sequence and a unique amino acid in the omega-site of the GPI anchor. To the best of our knowledge, this is the first report on genetic polymorphisms of the feline *SPRN* gene.

## 1. Introduction

Prion diseases are fatal and irreversible neurodegenerative disorders caused by a pathogenic isoform of prion protein (PrP^Sc^) changed from normal prion protein (PrP^C^) and have been reported in various hosts, including Creutzfeldt–Jakob disease (CJD), fatal familial insomnia (FFI) and Gerstmann–Sträussler–Scheinker syndrome (GSS) in humans; scrapie in sheep and goats; bovine spongiform encephalopathy (BSE) in cattle; transmissible mink encephalopathy (TME) in mink; chronic wasting disease in elk and deer; and feline spongiform encephalopathy (FSE) in cats, cheetahs and pumas [1,2,3,4,5,6,7,8]. Prion diseases are classified into three major types according to etiology, including sporadic, iatrogenic and genetic forms [9].

In recent studies, several cofactors that interact with prion protein (PrP), a template of PrP^Sc^, have been shown to accelerate the conversion of PrP^C^ to PrP^Sc^ [10,11,12,13]. Among these cofactors, the shadow of prion protein (Sho) encoded by the shadow of prion protein gene (*SPRN*) plays a pivotal role in pathogenesis in several types of prion diseases [14]. The Sho is a member of the prion protein family and shows similar characteristics to PrP, including being a glycosylphosphatidylinositol (GPI) anchor protein and having tandem repeat sequences [15]. Since the Sho is an important interaction partner for PrP, genetic polymorphisms in the *SPRN* gene that affect conformational changes and/or the expression level of the Sho protein are related to susceptibility to prion diseases in several hosts. In humans, a frameshift polymorphism in the *SPRN* gene is observed only in variant CJD patients [16]. In addition, insertion/deletion polymorphisms in the *SPRN* gene are associated with vulnerability to scrapie-affected goats and atypical BSE-affected cattle [17]. Although the role of the Sho protein in prion diseases is apparent, genetic polymorphisms in the feline *SPRN* gene and structural characteristics of the Sho have not yet been investigated in cats, a major host of FSEs.

In the present study, we performed amplicon sequencing to identify feline *SPRN* polymorphisms and analyzed the impact of feline *SPRN* polymorphisms on the secondary structure of *SPRN* gene mRNA using RNAsnp [18]. In addition, we performed multiple sequence alignments among several prion-related species to identify unique characteristics of the feline Sho protein using ClustalW [19]. Furthermore, we analyzed the N-terminal signal peptide and GPI-anchor using SignalP and PredGPI, respectively [20,21].

## 2. Materials and Methods

### 2.1. Ethical Statements

All experiments were performed according to the Korea Experimental Animal Protection Act. All experimental procedures were approved by the Institutional Animal Care and Use Committee (IACUC) of Jeonbuk National University (IACUC number: CBNU 2019-00077).

### 2.2. Samples

A total of 193 cat samples were provided by Hemalgeun and Cool-Pet animal hospitals in the Republic of Korea. These samples were derived from regular health examinations and emasculations performed by qualified veterinary surgeons. The tissue and blood samples originated from 17 cat breeds, including Korean Domestic Shorthair (162), Persian (8), Turkish Angora (3), Abyssinian (2), Bengal (2), Norwegian Forest (2), Russian Blue (2), Scottish Fold (2), Siamese (2), American Curl (1), American Shorthair (1), British Shorthair (1), Malaysian Domestic Shorthair (1), Minuet (also called Napoleon) (1), Siberian (1), Sphynx (1) and Ragdoll (1). Since the official pedigree document was not available, and the sample size of each breed was very small, breed-specific analysis was not conducted.

### 2.3. Genomic DNA Isolation

Genomic DNA was isolated from tissue and blood samples using the Labopass Tissue Genomic DNA Isolation Kit Mini (Cosmogenetech, Seoul, Korea) and Bead Genomic DNA Prep Kit for Blood (Biofact, Daejeon, Korea) following the manufacturers’ protocols.

### 2.4. Genetic Analysis

The feline *SPRN* gene (Gene ID: 102902356) was amplified from genomic DNA by polymerase chain reaction (PCR) using gene-specific primers (F: 5′-GCTGCGGTCCTTTCTCCGTT-3′, R: 5′-GCAGCGTCTGTGGGTCAG-3′). The PCR mixture was composed of 2.5 µL 10× H-star *Taq* reaction buffer, 5 µL 5× band helper, 1 µL dNTP mix (10 mM each dNTP), 1 µL of each primer (10 µM), 0.2 µL H-star *Taq* DNA polymerase (BIOFACT, Daejeon, Korea) and nuclease-free water up to a total volume of 25 µL. The PCR conditions were as follows: 98 °C for 15 min for denaturation; 40 cycles of 98 °C for 20 s, 60 °C for 40 s and 72 °C for 1 min for annealing and extension; and 1 cycle of 72 °C for 5 min for the final extension. PCR was performed using a C1000 Touch Thermal Cycler (Bio–Rad, Hercules, CA, USA). The PCR products (623 bp) were sequenced by an ABI 3730 sequencer (ABI, Forster City, CA, USA). Electropherograms of the sequencing data were produced by Finch TV software (Geospiza Inc., Seattle, WA, USA). The genotyping was performed with DNA sequences of the PCR products except for the primer region (585 bp).

### 2.5. Statistical Analyses

The Hardy–Weinberg equilibrium (HWE), linkage disequilibrium (LD) and haplotype analyses of the *SPRN* polymorphisms were carried out using Haploview version 4.2 (Broad Institute, Cambridge, MA, USA). Haplotypes were identified by in silico analysis using a haplotype reconstruction method [22]. Information about the *PRNP* polymorphisms was obtained from a previous study [8]. LD analysis between *PRNP* and *SPRN* polymorphisms was performed using Haploview version 4.2 (Broad Institute, Cambridge, MA, USA).

### 2.6. Multiple Sequence Alignments

The amino acid sequences of the Sho protein were obtained from GenBank at the National Center for Biotechnology Information (NCBI), including those of human (*Homo sapiens*, NP_001012526.2), mouse (*Mus musculus*, AAH56484.1), cattle (*Bos taurus*, AAY83885.1), sheep (*Ovis aries*, NP_001156033.1), goat (*Capra hircus*, AGU17009.1), red deer (*Cervus elaphus*, ACF24724.1), dog (*Canis lupus familiaris*, XP_038296952.1), horse (*Equus caballus*, XP_023492126.1) and cat (*Felis catus*, XP_023096939.1). The amino acid sequences of the Sho protein were aligned using ClustalW based on progressive alignment methods.

### 2.7. Secondary Structural Analysis of mRNA

‘Mode 1′ of RNAsnp (http://rth.dk/resources/rnasnp/) (accessed on 12 January 2022), which is specialized for short mRNA sequences (<1000 nt), was used to assess the effects of the feline *SPRN* polymorphisms on the mRNA structure according to RNA folding algorithms. The base-pair probabilities based on the global folding method were analyzed between wild-type and query RNA sequences. Structural differences between the wild-type and query RNA sequences were calculated using the Euclidean distance or Pearson correlation coefficient for all sequence intervals.

### 2.8. Prediction of the Signal Peptide of Feline Sho

The signal peptide and cleavage site of the Sho were predicted by SignalP 5.0 (https://services.healthtech.dtu.dk/service.php?SignalP-5.0) (accessed on 12 January 2022). The prediction of SignalP 5.0 was based on a deep neural network-based method combined with conditional random field classification and optimized transfer learning.

### 2.9. Prediction of Omega-Site and Signal Sequence of GPI-Anchor of Feline Sho

The omega-site and signal sequence of the GPI-anchor were predicted by PredGPI (http://gpcr.biocomp.unibo.it/predgpi/index.htm) (accessed on 12 January 2022). The anchoring signal was predicted based on a support vector machine (SVM). The omega-site was predicted based on a hidden Markov model (HMM).

## 3. Results

### 3.1. Identification of Novel Polymorphisms of the Feline SPRN Gene in 193 Cats

The feline *SPRN* gene is composed of two exons. To investigate feline *SPRN* gene polymorphisms, we performed PCR to amplify the open reading frame (ORF) region of the feline *SPRN* gene in Exon 2. The amplicons were composed of 623 bp, encompassing the ORF and a small part of the 3′ untranslated region (UTR) of the *SPRN* gene. We identified three novel single nucleotide polymorphisms (SNPs), including c.15C > A and c.430C > A in the ORF region and c.469C > T in the 3′ UTR (Figure 1). The genotype and allele frequencies of the SNPs in the feline *SPRN* gene are described in Table 1. The genotype frequencies of all polymorphisms were in HWE. We also investigated the LD values among the three feline *SPRN* polymorphisms with r^2^ values (Table 2). Notably, strong LD (r^2^ > 0.333) was not observed between any of the feline *SPRN* polymorphisms. In addition, we performed haplotype analysis of three polymorphisms in the feline *SPRN* gene (Table 3). The CCC haplotype was most frequently observed (63.5%), followed by the ACC (26.2%), CAC (10.1%) and CCT (3%) haplotypes in the feline *SPRN* gene (Table 3). Furthermore, we investigated LD values between feline *PRNP* and *SPRN* polymorphisms. Strong LD (r^2^ > 0.333) was not observed between feline *PRNP* and *SPRN* polymorphisms (Table 4).

### 3.2. mRNA Secondary Structure Analysis of the Feline SPRN Gene According to Polymorphisms

We investigated alterations of the mRNA secondary structure of the feline *SPRN* gene according to the haplotypes in the polymorphisms using the RNAsnp program (Figure 2). In the ACC haplotype in the feline *SPRN* polymorphisms, the graphical overview indicates a difference in the 1–215 nt region compared to the wild-type haplotype in the *SPRN* gene (CCC haplotype). The minimum free energies (CCC: −127.20 kcal/mol; ACC: −126.60 kcal/mol) and mRNA secondary structures showed differences between the CCC and ACC haplotypes (Figure 2A,B). In the CAC haplotype in the feline *SPRN* polymorphisms, the graphical overview indicates a difference in the 230–580 nt region compared to the wild-type haplotype in the *SPRN* gene (CCC haplotype). The minimum free energies (CCC: −206.00 kcal/mol; CAC: −205.90 kcal/mol) and mRNA secondary structures showed differences between the CCC and ACC haplotypes (Figure 2C,D). In the CCT haplotypes in the feline *SPRN* polymorphisms, the graphical overview indicates a difference in the 269–580 nt region compared to the wild-type haplotype in the *SPRN* gene (CCC haplotype). The minimum free energies (CCC: −176.10 kcal/mol; CCT: −175.20 kcal/mol) and mRNA secondary structures showed differences between the CCC and CCT haplotypes (Figure 2E,F).

### 3.3. Comparison of the Amino Acid Sequences of the Sho Protein among Species

We performed multiple sequence alignments of the amino acid sequences in the Sho protein among humans, mice, cattle, sheep, goats, red deer, dogs, horses and cats (Figure 3). The number of amino acids in the Sho protein was variable among species, and that of the human Sho protein was the longest (humans: 151; mice: 147; cattle: 143; sheep: 145; goats: 146; red deer: 143; dogs: 147; horses: 147; cats: 145). However, a low complexity Ala- and Gly-rich motif, in the interaction region of the Sho protein with PrP (red box), was conserved. In addition, although the NXT glycosylation motif (black box) was conserved among all species, cats have the unique amino acid sequence, NRT. The feline Sho protein showed a total of four feline-specific amino acids, including 21 I in the N-terminal domain, 106 R in the NXT glycosylation motif and 120 D and 138 V in the C-terminal domain.

### 3.4. Identification of Differences in the N-Terminal Signal Peptide of the Sho Protein among Species

We analyzed the N-terminal signal peptide of the Sho protein among humans, mice, cattle, sheep, goats, red deer, dogs, horses and cats. Detailed information on the amino acid sequences of the signal peptide is presented in Figure 4. The murine Sho has been reported to have an N-terminal signal sequence of 1–24 aa [23,24], and the prediction results in the present study were consistent with the data of the previous study. In brief, the length of the signal peptide of mice and dogs (24 aa) was longer than that of humans, cattle, sheep, goats, red deer, horses and cats (23 aa). Notably, the amino acid of the cutting site was alanine in all species.

### 3.5. Investigation of the Omega-Site and Signal Sequence of the GPl-Anchor of the Sho Protein

We analyzed the omega-site and signal sequences of the GPl anchor among humans, mice, cattle, sheep, goats, red deer, dogs, horses and cats (Table 5). The murine Sho has been reported to have a GPI signal sequence of 123–147 aa [23,25], and the prediction results (116–147 aa) of the present study showed a slight difference from the previous study. Mice and cattle had the longest signal sequences of the GPI anchor of the Sho protein (32 aa). In contrast, cats showed the shortest signal sequence of the GPI anchor of the Sho protein (27 aa). The amino acid of the omega-site of humans, sheep, goats, red deer, dogs and horses was serine. However, that of the mice and cattle was tyrosine. Notably, the amino acid of the omega-site of cats was aspartic acid.

## 4. Discussion

In the present study, we identified three novel SNPs in the feline *SPRN* gene (Figure 1, Table 1). We analyzed the effect of the three SNPs on the secondary structure of the mRNA of the *SPRN* gene using an in silico program. Notably, the secondary structure and minimum free energy of mRNAs were variable according to the haplotypes in the SNPs in the feline *SPRN* gene (Figure 2). In addition, SNPs can affect the transcription efficiency of genes according to how they affect the secondary structure of the mRNA. Thus, further analysis of the transcription level of the *SPRN* gene according to the haplotypes in the three SNPs is needed to validate the effects of the three SNPs. Among the three novel SNPs, c.469C > T was found in the 3′ UTR (Figure 1). The 3′ UTR is the transcription regulatory region that acts via the binding of microRNA [26,27]. Furthermore, a previous study reported that an insertion/deletion polymorphism in the caprine *SPRN* gene located on the 3′ UTR was associated with susceptibility to scrapie [28]. Since the expression level of the *SPRN* gene is important to the pathomechanism of prion diseases, further association analysis using FSE-infected animals and a reporter assay according to the genotype of c.469C > T is highly desirable in the future to validate the effect of c.469C > T.

To identify specific characteristics of the feline Sho protein, we compared the amino acid sequences of the Sho from several species. Notably, a low complexity Ala- and Gly-rich motif of the Sho was conserved among species. In addition, the feline Sho has a unique amino acid sequence: NRT in the NXT glycosylation motif (Figure 3). In addition, four feline-specific amino acids were found (Figure 3). One feline-specific amino acid, 106 R, was located in the NXT glycosylation motif, and the other three feline-specific amino acids were located on the signal sequences of the N-terminal signal peptide and GPI-anchor. Thus, we investigated the effect of feline-specific amino acids on the N-terminal signal peptide and the signal sequence and omega-site of the GPI-anchor. In the N-terminal signal peptide, the cutting site and length of the signal sequence of cats showed similarity with those of humans, cattle, sheep, goats, red deer and horses (Figure 4). This in silico result may suggest that the feline-specific amino acid 21I did not significantly affect the interspecific conserved signal sequence or cutting site of the N-terminal signal peptide. However, the feline Sho protein showed the shortest signal sequence and a unique amino acid in the omega-site of the GPI anchor (Table 5). This in silico result may suggest that the feline-specific amino acids 120 D and 138 V significantly affect the interspecific conserved signal sequence and omega-site of the GPI-anchor. However, since these in silico analyses were the prediction, further validation of the N-terminal signal sequence and omega site of GPI-anchor is highly desirable using biochemical analyses. In recent studies, the Sho protein has been shown to have a stress-resistant function and to be essential for early mouse embryogenesis and mammary development and differentiation [6,7,29]. Future analysis of the effect of GPI-anchor-related differences on the native function and physiology of the Sho protein is greatly desired.

## 5. Conclusions

In conclusion, we identified three novel SNPs in the feline *SPRN* gene. Strong LD was not observed among the three SNPs. We also found four major haplotypes in the *SPRN* polymorphisms. Strong LD was not observed between *PRNP* and *SPRN* polymorphisms. In addition, we found different secondary structures and minimum free energies of mRNA according to the haplotypes in the *SPRN* polymorphisms. Furthermore, we found four feline-specific amino acids in the feline Sho using multiple sequence alignments among several species. Lastly, the signal sequence and the cutting site of the N-terminal signal peptide of the Sho protein in cats showed similarity with those in humans, cattle, sheep, goats, red deer and horses. However, the feline Sho protein showed the shortest signal sequence and a unique amino acid in the omega-site of the GPI anchor. To the best of our knowledge, this is the first report on genetic polymorphisms in the feline *SPRN* gene.

## Figures and Tables

**Figure 1 viruses-14-00981-f001:**
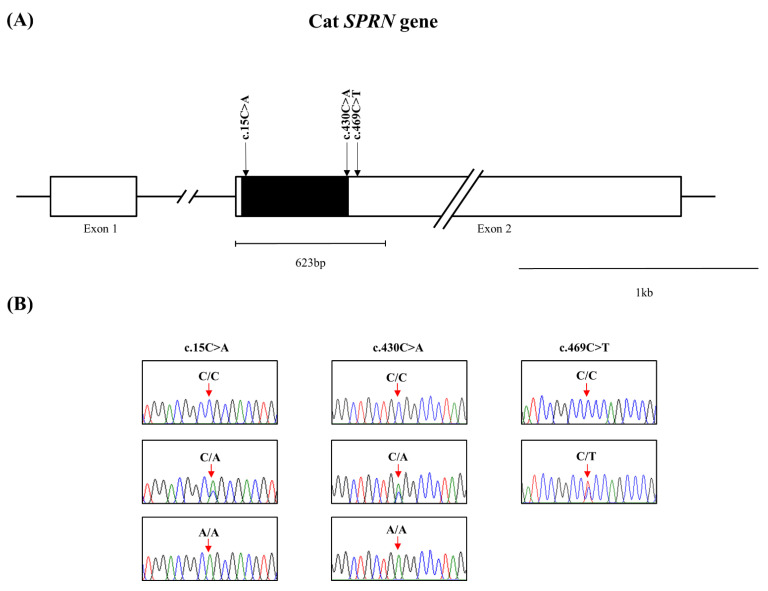
Novel genetic polymorphisms in the feline shadow of prion protein gene (*SPRN*) in the present study. (**A**) Gene map of polymorphisms in the feline *SPRN* gene on chromosome D2. The open reading frame (ORF) is indicated by a shaded block, and the 5′ and 3′ untranslated regions (UTRs) are indicated by white blocks. The edged horizontal bar indicates the regions sequenced. (**B**) Electropherograms of 3 novel single nucleotide polymorphisms (SNPs) in the feline shadow of prion protein gene (*SPRN*) found in cats. The four colors of the peaks indicate each base of the DNA sequence (green: adenine; red: thymine; blue: cytosine; black: guanine).

**Figure 2 viruses-14-00981-f002:**
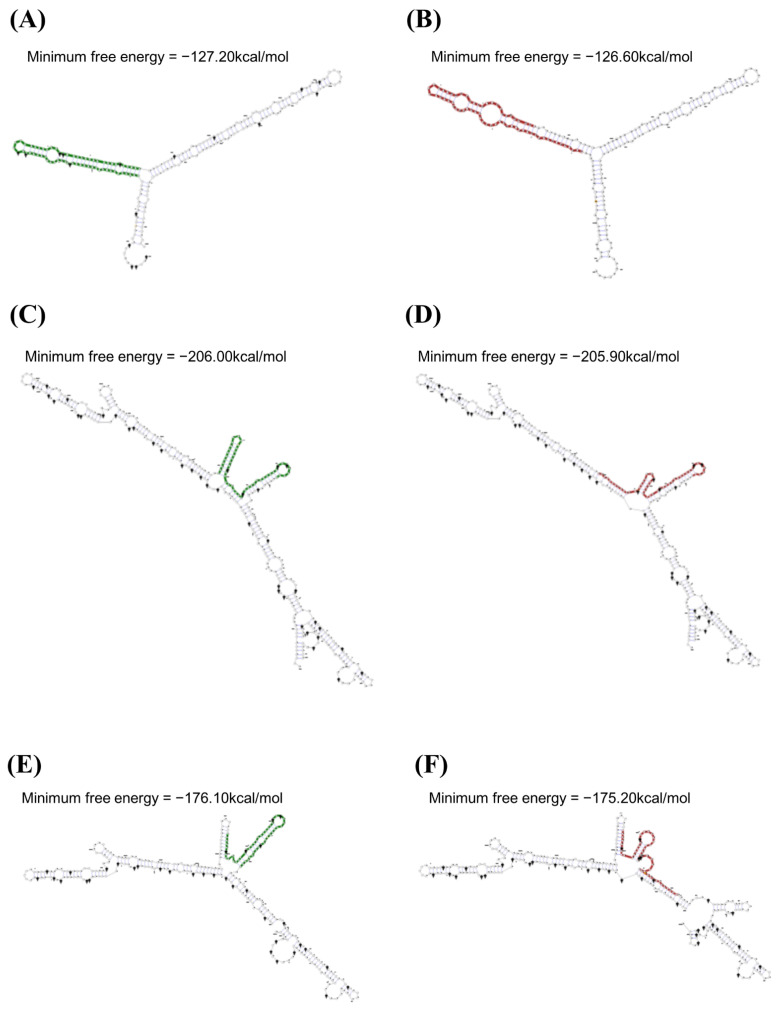
Analysis of the mRNA secondary structure of the feline *SPRN* gene according to haplotypes in the feline *SPRN* polymorphisms. (**A**) Minimum free energy and mRNA secondary structure of the feline *SPRN* gene with the CCC haplotype (1–215 nt). (**B**) Minimum free energy and mRNA secondary structure of the feline *SPRN* gene with the ACC haplotype (1–215 nt). (**C**) Minimum free energy and mRNA secondary structure of the feline *SPRN* gene with the CCC haplotype (230–580 nt). (**D**) Minimum free energy and mRNA secondary structure of the feline *SPRN* gene with the CAC haplotype (230–580 nt). (**E**) Minimum free energy and mRNA secondary structure of the feline *SPRN* gene with the CCC haplotype (269–580 nt). (**F**) Minimum free energy and mRNA secondary structure of the feline *SPRN* gene with the CCT haplotype (269–580 nt).

**Figure 3 viruses-14-00981-f003:**
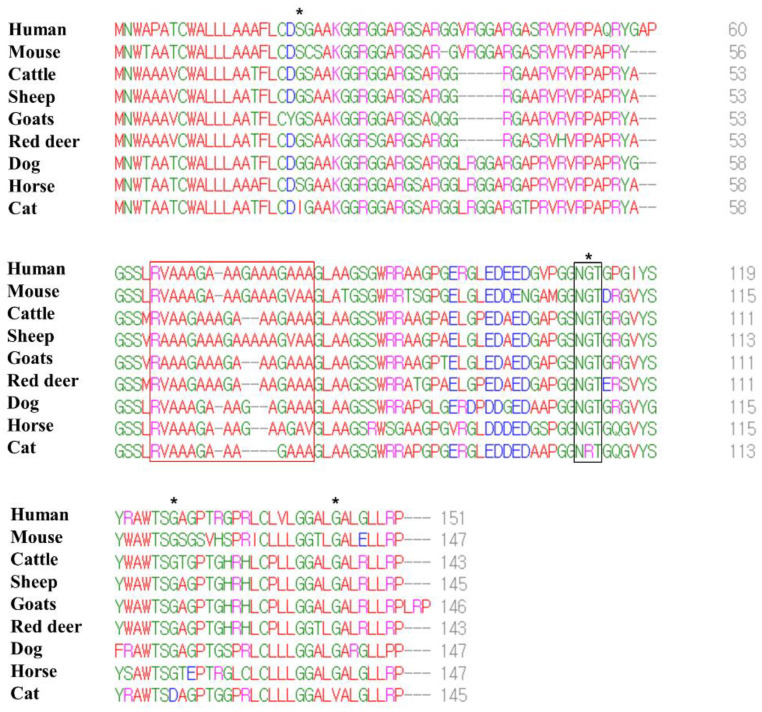
Multiple sequence alignments of the amino acid sequences in the shadow of prion protein (Sho) in humans, mice, cattle, sheep, goats, red deer, dogs, horses and cats. Colors indicate the chemical properties of the amino acids; blue: acidic; red: small and hydrophobic; magenta: basic; green: hydroxyl, sulfhydryl, amine and glycine. Asterisks indicate cat-specific residues. The red box indicates the interaction region of the Sho protein with prion protein (PrP). The black box indicates the NXT glycosylation motif of the Sho protein.

**Figure 4 viruses-14-00981-f004:**
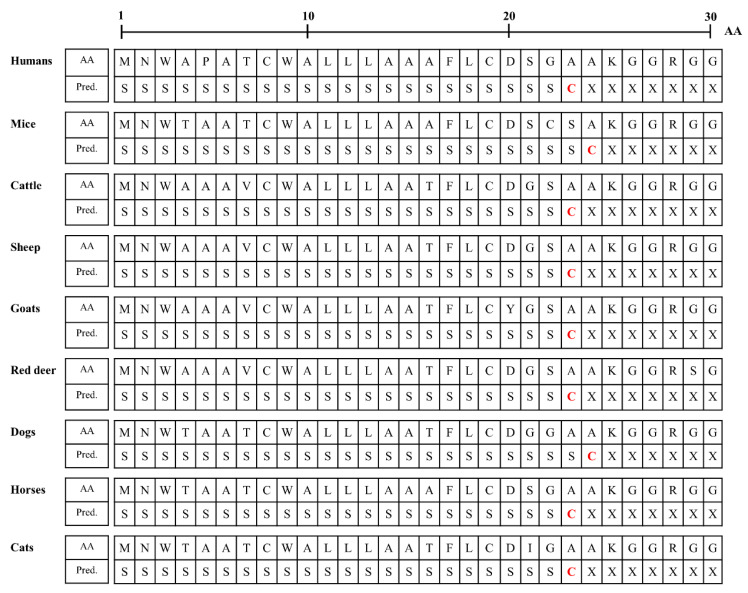
Signal peptide predictions of amino acid sequences of Sho protein in human, mouse, cattle, sheep, goats, red deer, dog, horse and cat. AA: amino acids; Pred.: prediction; S: signal peptide; C: cutting site; X: non-signal peptide.

**Table 1 viruses-14-00981-t001:** Genotype and allele frequencies of shadow of prion protein gene (*SPRN*) polymorphisms in 193 domestic cats.

Polymorphisms	Genotype Frequency, *n* (%)			Allele Frequency, *n* (%)		HWE
c.15C > A	CC 109 (56.48)	CA 67 (34.72)	AA 17 (8.81)	C 285 (73.83)	A 101 (26.17)	0.158
c.430C > A	CC 157 (81.35)	CA 33 (17.10)	AA 3 (1.55)	C 346 (89.90)	A 39 (10.10)	0.414
c.469C > T	CC 192 (99.48)	CT 1 (0.52)	TT 0 (0)	C 385 (99.74)	T 1 (0.26)	0.971

HWE: Hardy-Weinberg Equilibrium.

**Table 2 viruses-14-00981-t002:** Linkage disequilibrium (LD) among genetic polymorphisms in the *SPRN* gene in cats.

r2	c.15C > A	c.430C > A	c.469C > T
c.15C > A	-		
c.430C > A	0.04	-	
c.469C > T	0.001	0	-

**Table 3 viruses-14-00981-t003:** Haplotype frequencies of 3 *SPRN* polymorphisms in cats.

Haplotype	c.15C > A	c.430C > A	c.469C > T	Frequency (*n* = 386)
ht1	C	C	C	245 (0.635)
ht2	A	C	C	101 (0.262)
ht3	C	A	C	39 (0.101)
ht4	C	C	T	1 (0.003)

**Table 4 viruses-14-00981-t004:** LD between polymorphisms in the *PRNP* and *SPRN* genes with r^2^ values in cats.

r^2^	*SPRN* c.15C > A	*SPRN* c.430C > A	*SPRN* c.469C > T
*PRNP* c.-3G > A	0.002	0.007	0.0
*PRNP* c.128G > A	0.007	0.0	0.0
*PRNP* c.171C > T	0.001	0.007	0.002
*PRNP* c.201C > T	0.003	0.004	0.005
*PRNP* c.214_240del CCCCACGCC GGCGGAGGC TGGGGTCAG	0.007	0.03	0.0
*PRNP* c.255T > C, G	0.0	0.004	0.032
*PRNP* c.264T > C	0.0	0.008	0.015
*PRNP* c.279C > T	0.0	0.0	0.0
*PRNP* c.457G > A	0.003	0.001	0.023
*PRNP* c.734C > T	0.003	0.007	0.0
*PRNP* c.774C > T	0.0	0.002	0.0
*PRNP* c.787C > T	0.002	0.046	0.0
*PRNP* c.789G > A	0.004	0.009	0.0
*PRNP* c.790C > T	0.0	0.011	0.0
*PRNP* c.979G > A	0.002	0.007	0.0

**Table 5 viruses-14-00981-t005:** Prediction of the omega-site and signal sequences of the glycosylphosphatidylinositol (GPI)-anchor of the Sho protein by PredGPI.

Species	Omega-Site		Signal Sequence		
	Position	Amino Acid	Position	Length	Protein Sequence
Humans	125	S	125–151	27	SGAGPTRGPRLCLVLGGALGALGLLRP
Mice	116	Y	116–147	32	YWAWTSGSGSVHSPRICLLLGGTLGALELLRP
Cattle	112	Y	112–143	32	YWAWTSGTGPTGHRHLCPLLGGALGALRLLRP
Sheep	119	S	119–145	27	SGAGPTGHRHLCPLLGGALGALRLLRP
Goats	117	S	117–146	30	SGAGPTGHRHLCPLLGGALGALRLLRPLRP
Red deer	117	S	117–143	27	SGAGPTGHRHLCPLLGGTLGALRLLRP
Dogs	121	S	121–147	27	SGAGPTGSPRLCLLLGGALGARGLLPP
Horses	117	S	117–147	31	SAWTSGTEPTRGLCLCLLLGGALGALGLLRP
Cats	120	D	120–145	26	DAGPTGGPRLCLLLGGALVALGLLRP

## Data Availability

The data presented in this study are available on request from the corresponding author.

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
