# Peer review of "Novel Polymorphisms and Genetic Characteristics of the Shadow of Prion Protein Gene (SPRN) in Cats, Hosts of Feline Spongiform Encephalopathy"

_viruses, 2022, doi:10.3390/v14050981_

Round 1
Reviewer 1 Report
I reviewed the article titled "Novel Polymorphisms and Genetic Characteristics of the 2 Shadow of Prion Protein Gene (SPRN) in Cats, Hosts of Feline 3 Spongiform Encephalopathy".
The authors of the paper sequenced the SPRN gene that may be involved in prion disease. They analyzed the SPRN gene of 193 cats and identified 3 new variants. They also found 4 cat-specific amino acids. They report that their work is the first report of genetic polymorphisms of the feline SPRN gene.
The work is well designed and the manuscript is well written.
I have a small suggestion for the material methods part.
The length of the PCR product and the sequence length can be given.
Kind regards
Author Response
Reviewer 1
- The length of the PCR product and the sequence length can be given.
Response: Thank for the reviewer’s good comment. We have added the information on the length of the PCR product and the DNA sequence in the Materials and methods section [Page 8, lines 12, 15-16].
Reviewer 2 Report
In this study the authors focus on the analysis of genetic polymorphisms in the feline SPRN gene. Considering that cats are common pets in human households, and that Sho interacts with PrP and genetic polymorphisms in SPRN affect susceptibility to prion diseases, the study is definitely worthwhile. The fact that very little research has been done on feline PrP and SPRN further underscores the importance of the work. The authors identify 3 novel SNPs in the feline SPRN and 4 SPRN haplotypes. They then analyze potential changes in the secondary mRNA structure due to these polymorphisms in silico. In another line of research, they identify 4 feline-specific amino acids in the Sho protein, including a unique amino acid in the GPI anchor site and a potentially inactivating change in the NXT glycosylation motif. The major weakness of the paper is that the in-silico-based hypotheses are not substantiated by further biochemical experiments. As the authors acknowledge in the discussion, it remains unclear if the SNPs they identify have any effect on the expression of SPRN, and on the susceptibility to FSE. Analogously, it remains unclear if the feline Sho is different from Sho’s from other species in terms of glycosylation or GPI anchoring. However, I believe that the manuscript can be published with the current scope of work with a hope that further studies will come from this or other labs. Thus, I would recommend acceptance after major textual revisions. The only real concern is the Figure 2 legend: if the authors just made 6 same-word errors in this legend, it can be corrected. If they somehow indeed work with the equine SPRN sequence, the data cannot be examined without a detailed explanation on the similarity of the equine and feline SPRNs and should be reviewed again after the authors provide all the details.
Major issues:
Nomenclature: Throughout manuscript the correct nomenclature for PrPSc and PrPC requires superscript (rather than just PrPSC and PrPC).
The authors only sequence a 623 bp long region of SPRN encompassing the ORF and just a small part of the 3’ UTR in the second exon. Neither the first exon, nor the end of the second exon are analyzed. This should be very clearly stated in the Abstract and in the first section of the Results. While the ORF is more interesting, the fact that not the entire mRNA sequence is analyzed may be important.
It would be interesting to see how frequently the polymorphisms are seen in different breeds. Considering that 162 out of 193 samples come from one breed, does this particular breed carry all the polymorphisms? Or do the polymorphisms a breed-specific? If the authors deem breed determination not reliable and do not feel comfortable to analyze this aspect, it is worth mentioning this in Materials and Methods.
The 469C>T mutation has only been detected in 1 sample (HWE=0.971). Is it appropriate to include this polymorphism in the LD statistical analysis? In my opinion, it cannot be done when the lowest number is 1.
Table 3. Please, clarify, how the frequency of haplotypes was determined. This is not obvious. Chromatograms provided would not allow to determine a haplotype. Specifically, how the haplotypes would be determined for, say, a sample that has a C/A for 15C>A and a C/A for 430C>A? Both CC /AA, and CA / AC are possible. Based on the lack of the AAC haplotype in Table 3, one can understand that this is just a hypothetical situation, but this should be explained clearly in order to not make the reader do the math.
Table 4. The description of the analysis of PrP sequences is totally missing in Materials and Methods and there is no legend for the Table. Is it possible that Table legends had gone missing during the submission altogether?
Figure 2. Why is equine mRNA SPRN structures are used throughout Figure 2, while the text in the Results says that the structures are feline? Are these this just 6 typos? Then, please, correct. If the authors actually superimpose the polymorphisms on the already known equine SPRN mRNA structure, this is a major problem and may determine the acceptance of the manuscript. In my opinion, the data can only be presented with a detailed explanation of ALL other changes in these regions between the feline and equine SPRN genes, and may not be reliable at all.
Lines 189-191. In my opinion, it is not correct to say that the SEQUENCE of the PrP-binding region is conserved. Actually, this is one of the most variable regions of the Sho protein in all species. It would be correct to say that a low complexity Ala- and Gly-rich motif is conserved. Also, while the legend says that “The red box indicates the interaction region of the Sho protein and its conservation with that of the prion protein (PrP)”, the PrP sequence is not provided. Please, include the PrP sequence (obviously, feline), or revise the figure legend.
Line 190. While it is correct to say that the NXT glycosylation motif is conserved among species, it is definitely NOT conserved in cats. And this should be underscored. The change of G to R may totally block glycosylation.
Figure 4. Without references it is hard to understand which evidence for the signal peptide is experimentally confirmed, and which is based on predictions only. This would help to evaluate the validity of in-silico predictions for cats.
Figure 5. Without a reference, it is not clear if the GPI anchor site in cats is functional, and if its functionality is similar to other species. The change to aspartic acid in the omega position is pretty dramatic.
Lines 238-239. The authors say :” To identify specific characteristics of the feline Sho protein, we compared the amino acid sequences of PrPs among several species. The interaction region and the NXT glycosylation motif of the feline Sho protein are conserved with those of PrP.” However, in the Result section they only compare the SPRN sequences of different species and there the NXT glycosylation site is not conserved. Please, clarify and either provide a reference or, better yet, a sequence comparison.
Lines 244-245. The authors say “…we investigated the effect of feline-specific amino acids on the N-terminal signal peptide …” and go on to conclude that “the feline-specific amino acid 21I did not significantly affect the interspecific conserved signal sequence or cutting site of the N-terminal signal peptide”. I honestly think that this is a huge overstatement without a biochemical confirmation. At most, their in silico analysis in silico data may suggest such a conclusion. I would recommend to revise the text.
The same relates to the discussion of the GPI anchor site – without biochemical evidence it is premature
Minor issues:
Lines 26-27. “In addition, we found alterations in the secondary structure and minimum free energy of mRNA according to the haplotypes of the SPRN polymorphisms, and we found 4 feline-specific amino acids ……” Because the sentence describes 2 different approaches to the analysis to SPRN / Sho, I suggest to break the sentence in two. The same relates to Conclusions line 261-264.
Lines 36-41. Considering that the study singles out one of multiple species for genetic analysis, to make the list of susceptible species more complete, consider adding chronic wasting disease in elk and deer and mink spongiform encephalopathy.
Introduction. Some readers from outside the prion field (i.e. focusing on polymorphism analysis) may benefit from at least a 1-sentence description of the unique mechanism of prion formation and propagation stressing that prion diseases may be sporadic, infections and genetically pre-determined.
Line 99. Please, substitute the real reference for a [Ref].
Line 114. “The structural differences between wild-type and query RNA sequences was calculated…” Please, correct grammar.
Lines 117 and 122. I would suggest adding the word “feline” into the section title. The existence a signal peptide and GPI anchor sites for other species is already known.
Lines 132-133. “… and c.469C>T in the 3ʹ untranslated regions (UTRs) (Figure 1). “ Please, consider revising – if only one UTR polymorphism was found, then, I assume, one should say “…and c.469C>T in the 3ʹ untranslated region (UTR) (Figure 1)”.
Line 213. There is no section title indicating the analysis of the GPI anchor site. Please, either add it to the previous section title, or make a separate one.

Author Response
Reviewer 2
- Nomenclature: Throughout manuscript the correct nomenclature for PrPSc and PrPC requires superscript (rather than just PrPSC and PrPC).
Response: Thank for the reviewer’s good comment. We have corrected nomenclature for PrPSc and PrPC throughout the manuscript.
- The authors only sequence a 623 bp long region of SPRN encompassing the ORF and just a small part of the 3’ UTR in the second exon. Neither the first exon, nor the end of the second exon are analyzed. This should be very clearly stated in the Abstract and in the first section of the Results. While the ORF is more interesting, the fact that not the entire mRNA sequence is analyzed may be important.
Response: Thank for the reviewer’s good comment. The information on the PCR products analyzed in this study has been added in the Abstract section [Page 3, lines 10-11] and Results section [Page 11, lines 5-6].
- It would be interesting to see how frequently the polymorphisms are seen in different breeds. Considering that 162 out of 193 samples come from one breed, does this particular breed carry all the polymorphisms? Or do the polymorphisms a breed-specific? If the authors deem breed determination not reliable and do not feel comfortable to analyze this aspect, it is worth mentioning this in Materials and Methods.
Response: Thank for the reviewer’s good comment. Since the official pedigree document was not available and the sample size of each breed was very small, we could not perform a breed-specific analysis. This information has been added in the Materials and methods section [Page 7, lines 15-16].
- The 469C>T mutation has only been detected in 1 sample (HWE=0.971). Is it appropriate to include this polymorphism in the LD statistical analysis? In my opinion, it cannot be done when the lowest number is 1.
Response: Thank for the reviewer’s good comment. According to the reviewer’s suggestion, we have omitted the results regarding 469C>T in the Table 3.
- Table 3. Please, clarify, how the frequency of haplotypes was determined. This is not obvious. Chromatograms provided would not allow to determine a haplotype. Specifically, how the haplotypes would be determined for, say, a sample that has a C/A for 15C>A and a C/A for 430C>A? Both CC /AA, and CA / AC are possible. Based on the lack of the AAC haplotype in Table 3, one can understand that this is just a hypothetical situation, but this should be explained clearly in order to not make the reader do the math.
Response: Thank for the reviewer’s good comment. We have added the information on haplotype analysis in the Materials and methods section [Page 8, lines 21-22] and References section [Page 22 lines 14-15].
- Table 4. The description of the analysis of PrP sequences is totally missing in Materials and Methods and there is no legend for the Table. Is it possible that Table legends had gone missing during the submission altogether?
Response: Thank for the reviewer’s good comment. As suggested by the reviewer, we have added the information on the analysis of PrP sequences in the Materials and methods section [Page 9, lines 1-2] and Table 4.
- Figure 2. Why is equine mRNA SPRN structures are used throughout Figure 2, while the text in the Results says that the structures are feline? Are these this just 6 typos? Then, please, correct. If the authors actually superimpose the polymorphisms on the already known equine SPRN mRNA structure, this is a major problem and may determine the acceptance of the manuscript. In my opinion, the data can only be presented with a detailed explanation of ALL other changes in these regions between the feline and equine SPRN genes, and may not be reliable at all.
Response: Thank for the reviewer’s good comment. We made a typo mistake in legend of the Figure 2. As suggested by the reviewer, we have changed the “equine” to “feline” in the Figure legends section [Page 17, lines 13-15, 17-18, 20].
- Lines 189-191. In my opinion, it is not correct to say that the SEQUENCE of the PrP-binding region is conserved. Actually, this is one of the most variable regions of the Sho protein in all species. It would be correct to say that a low complexity Ala- and Gly-rich motif is conserved.
Response: Thank for the reviewer’s good comment. As suggested by the reviewer, we have modified the sentences in the Results section [Page 12, lines 19-20] and Discussion section [Page 14, lines 18-19].
- Also, while the legend says that “The red box indicates the interaction region of the Sho protein and its conservation with that of the prion protein (PrP)”, the PrP sequence is not provided. Please, include the PrP sequence (obviously, feline), or revise the figure legend.
Response: Thank for the reviewer’s good comment. We have modified the sentences in the Figure legends section [Page 18, lines 3-4].
- Line 190. While it is correct to say that the NXT glycosylation motif is conserved among species, it is definitely NOT conserved in cats. And this should be underscored. The change of G to R may totally block glycosylation.
Response: Thank for the reviewer’s good comment. As suggested by the reviewer, we have modified the sentences in the Results section [Page 12, lines 20-22] and Discussion section [Page 14, lines 19-20].
- Figure 4. Without references it is hard to understand which evidence for the signal peptide is experimentally confirmed, and which is based on predictions only. This would help to evaluate the validity of in-silico predictions for cats.
Response: Thank for the reviewer’s good comment. As suggested by the reviewer, we have added the sentences in the Results section [Page 13, lines 6-8] and references in the References section [Page 22, lines 16-22].
- Figure 5. Without a reference, it is not clear if the GPI anchor site in cats is functional, and if its functionality is similar to other species. The change to aspartic acid in the omega position is pretty dramatic.
Response: Thank for the reviewer’s good comment. As suggested by the reviewer, we have added the sentences in the Results section [Page 13, lines 16-18] and references in the References section [Page 22, lines 16-20, 23-25; Page 23, line 1].
- Lines 238-239. The authors say :” To identify specific characteristics of the feline Sho protein, we compared the amino acid sequences of PrPs among several species. The interaction region and the NXT glycosylation motif of the feline Sho protein are conserved with those of PrP.” However, in the Result section they only compare the SPRN sequences of different species and there the NXT glycosylation site is not conserved. Please, clarify and either provide a reference or, better yet, a sequence comparison.
Response: Thank for the reviewer’s good comment. As suggested by the reviewer, we have modified the sentences in the Discussion section [Page 14, lines 18-20].
- Lines 244-245. The authors say “…we investigated the effect of feline-specific amino acids on the N-terminal signal peptide …” and go on to conclude that “the feline-specific amino acid 21I did not significantly affect the interspecific conserved signal sequence or cutting site of the N-terminal signal peptide”. I honestly think that this is a huge overstatement without a biochemical confirmation. At most, their in silico analysis in silico data may suggest such a conclusion. I would recommend to revise the text.
Response: Thank for the reviewer’s good comment. As suggested by the reviewer, we have modified the sentences in the Discussion section [Page 15, lines 3, 8-10].
- The same relates to the discussion of the GPI anchor site – without biochemical evidence it is premature.
Response: Thank for the reviewer’s good comment. As suggested by the reviewer, we have modified the sentences in the Discussion section [Page 15, lines 6-10].
- Lines 26-27. “In addition, we found alterations in the secondary structure and minimum free energy of mRNA according to the haplotypes of the SPRN polymorphisms, and we found 4 feline-specific amino acids ……” Because the sentence describes 2 different approaches to the analysis to SPRN / Sho, I suggest to break the sentence in two. The same relates to Conclusions lines 261-264.
Response: Thank for the reviewer’s good comment. As suggested by the reviewer, we have modified the sentences in the Abstract section [Page 3, lines 18-21] and Conclusions section [Page 16, lines 4-7].
- Lines 36-41. Considering that the study singles out one of multiple species for genetic analysis, to make the list of susceptible species more complete, consider adding chronic wasting disease in elk and deer and mink spongiform encephalopathy.
Response: Thank for the reviewer’s good comment. As suggested by the reviewer, we have added the sentences in the Introduction section [Page 5, lines 6-7].
- Introduction. Some readers from outside the prion field (i.e. focusing on polymorphism analysis) may benefit from at least a 1-sentence description of the unique mechanism of prion formation and propagation stressing that prion diseases may be sporadic, infections and genetically pre-determined.
Response: Thank for the reviewer’s good comment. As suggested by the reviewer, we have added the sentences in the Introduction section [Page 5, lines 8-10].
- Line 99. Please, substitute the real reference for a [Ref].
Response: Thank for the reviewer’s good comment. As suggested by the reviewer, we have added the reference in the Materials and methods section [Page 8, line 22] and References section [Page 20, lines 23-25; Page 21, line 1-2].
- Line 114. “The structural differences between wild-type and query RNA sequences was calculated…” Please, correct grammar.
Response: Thank for the reviewer’s good comment. As suggested by the reviewer, we have modified the sentences in the Materials and methods section [Page 9, line 20].
- Lines 117 and 122. I would suggest adding the word “feline” into the section title. The existence a signal peptide and GPI anchor sites for other species is already known.
Response: Thank for the reviewer’s good comment. As suggested by the reviewer, we have added “feline” in the Materials and methods section [Page 10, line 1; Page 10, line 7].
- Lines 132-133. “… and c.469C>T in the 3ʹ untranslated regions (UTRs) (Figure 1). “ Please, consider revising – if only one UTR polymorphism was found, then, I assume, one should say “…and c.469C>T in the 3ʹ untranslated region (UTR) (Figure 1)”.
Response: Thank for the reviewer’s good comment. As suggested by the reviewer, we have changed “UTRs” to “UTR” in the Results section [Page 11, line 8].
- Line 213. There is no section title indicating the analysis of the GPI anchor site. Please, either add it to the previous section title, or make a separate one.
Response: Thank for the reviewer’s good comment. As suggested by the reviewer, we have added the section title in the Results section [Page 13, lines 13-14].
Round 2
Reviewer 2 Report
The authors did a very good job on revising the manuscript. All the Reviewer’s comments were addressed. I recommend acceptance.
There are just couple very minor issues:
Page 8 lines 21-22. In the sentence “Haplotypes were performed by in silico analysis using reconstruct haplotypes method [22]”, the word “performed” is not appropriate. I guess, the haplotypes were “identified”?
Have the newly identified sequence polymorphisms been downloaded to a publicly available database, e.g. NCBI. I am not 100% sure about Viruses, but most journals require this statement, and the access numbers.
Author Response
Reviewer 2
- Page 8 lines 21-22. In the sentence “Haplotypes were performed by in silico analysis using reconstruct haplotypes method [22]”, the word “performed” is not appropriate. I guess, the haplotypes were “identified”?
Response: Thank for the reviewer’s good comment. We have changed “performed” to “identified” in the Materials and methods section [Page 8, line 21].
- Have the newly identified sequence polymorphisms been downloaded to a publicly available database, e.g. NCBI. I am not 100% sure about Viruses, but most journals require this statement, and the access numbers.
Response: Thank for the reviewer’s good comment. We found novel genetic polymorphisms of the feline SPRN gene for the first time in the present study, and the registration of these polymorphisms is in progress at the NCBI.